# Drain-Source Voltage-Controllable Three-Switch Active-Clamp Forward Converter for Wide Input/Output Voltage Applications

**DOI:** 10.3390/mi14010035

**Published:** 2022-12-23

**Authors:** Jeong-Woo Lim, Chong-Eun Kim

**Affiliations:** Department of Control and Instrumentation Engineering, Gyeongsang National University, Jinju 52828, Republic of Korea

**Keywords:** active clamp, forward converter, three switch

## Abstract

Active-clamp forward converters are applied to various medium-capacity power systems because they have a relatively simple structure and are capable of zero-voltage switching. In particular, there is the advantage that a stable output voltage can be obtained by controlling the duty ratio of the power semiconductor switch even in applications with wide input and output voltage ranges. However, the voltage stress on the power semiconductor switches due to the application of active clamp is higher than the input voltage, especially as the duty ratio increases. A three-switch active-clamp forward converter is proposed, which can overcome such shortcomings and can reduce the voltage stress of the power semiconductor switches, but it causes an increase in the DC bias of the magnetizing current and the additional conduction and switching losses. Therefore, in this paper, a voltage-stress-controllable three-switch active-clamp forward converter that can utilize both advantages of the conventional active-clamp forward converter and three-switch active clamp forward converter is proposed and verified through a prototype for 800 W battery charger.

## 1. Introduction

In recent years, with the development of various mobile devices, QC (Quick Charge) and PD (Power Delivery) chargers that can rapidly charge various types of batteries have been released, and the maximum output voltage and power capacity are increasing [1,2]. In addition, as smart mobility services expand, demand of battery charging systems for electric scooters is increasing, and battery sharing services for electric motorcycles are also expanding from Southeast Asia [3,4,5]. Furthermore, the spread of electric vehicles is expanding due to the de-petroleum and carbon-neutral policies that are being implemented all around the world. Accordingly, the demands for driving distance and large-capacity batteries are increasing. That is, active research and development of power converters for battery-powered systems is required [6,7,8].

Since the battery has a wide voltage range, the power system for the battery has a wide input voltage range, and the battery charging system must cover the wide output voltage range [9,10]. Therefore, a power conversion topology with a wide input/output voltage gain is needed. For a low power rating of less than 100 W, the active-clamp flyback converter is mainly used, and the LLC resonant converter and phase-shift full-bridge converter are applied as a high-power topology of 1 kW or above. However, the LLC resonant converter is difficult to use as a battery charger due to disadvantages such as reduced efficiency and increased price as it is applied in combination with a pre-regulator or post-regulator depending on the narrow input/output voltage gain [11,12,13,14,15]. Moreover, a highly reliable and robust control method may be required to commercialize these topologies. However, in the case of a powerful control method, implementation is very complicated, and it is difficult to apply such a complex control method to actual products [16,17].

On the other hand, the active-clamp forward converter in Figure 1a with a wide input/output voltage gain according to the change of the duty ratio can be applied [18,19,20,21,22]. Continuous current is supplied thanks to the output inductor, and power efficiency can be improved through zero-voltage switching. However, the voltage stress of the power semiconductor switch is higher than the input voltage and, in particular, increasing voltage stress due to increased duty ratio is a drawback. As shown in Figure 1b, a three-switch active-clamp forward converter with additional one switch and one diode has been proposed to reduce the voltage stress of the power semiconductor switch [23,24].

Nevertheless, the voltage applied to the transformer increases due to the added switch, and as the magnetizing current is biased to the positive direction, the peak current increases and the zero-voltage switching performance is deteriorated. Therefore, in this paper, by varying the duty of the additional switch *Q_T_*, according to the load conditions. The voltage-stress-controllable three-switch active-clamp forward converter, which can take all the advantages by properly switching between the operation of the conventional active clamp forward converter and three-switch active clamp forward converter, is proposed and verified through a prototype circuit for the 800 W battery charger.

## 2. Analysis of Voltage Stress

The voltage stress of the main switch in the forward converter topology with the active clamp circuit can be expressed as the sum of the input voltage and the clamp capacitor voltage. In this case, it shows that the voltage of the clamp capacitor increases when the duty ratio increases as shown in Figure 2a. Additionally, in the case of a three-switch active clamp-forward converter, the voltage stresses of the main switch, *Q_M_* and clamp switch, *Q_C_* can be obtained by the sum of the input voltage and the voltage of the clamp capacitor.

However, with the additional switch *Q_T_*, as shown in Figure 3, both the input voltage and the clamp capacitor voltage contribute to the transformer reset operation, which indicates that the voltage of the clamp capacitor is lower than that of the conventional active clamp-forward converter. Therefore, it can be confirmed that the voltage stress of the three-switch active clamp-forward converter is relatively lower than that of the conventional active clamp-forward converter as shown in Figure 2b. In addition, conventional active clamp forward converters achieve zero-voltage switching from the application of active clamp circuits, and the voltage charged to clamp capacitor *C_C_* is added to the input voltage to determine the voltage stress of primary main switch *Q_M_* and clamp switch *Q_C_*. From the voltage second balance equation for transformer *T_F_*, voltage stress of clamp capacitor *V_Cc_* and voltage stress of main switch *Q_M_* and clamp switch *Q_C_* as a function of *D* are induced as shown in Equations (1)–(3).
(1)$$DV_{in}-\left(1-D\right)V_{Cc}=0$$
(2)$$V_{Cc}=\frac{D}{1-D}V_{in}$$
(3)$$V_{DS.max}\left(Q_{M}\;and\;Q_{C}\right)=\frac{V_{in}}{1-D}$$

On the other hand, in the case of a three-switch active-clamp forward converter, the additional switch *Q_T_* turns on and off at the same time as the main switch *Q_M_*, so the voltage charged in the clamp capacitor *C_C_* is relatively reduced. The voltage stresses of *Q_M_* and *Q_C_* are formed by the voltage second balance equation for transformer *T_F_*_,_ and it can be derived as the following Equations (4)–(8).
(4)$$DV_{in}-D_{1}\left(V_{in}+V_{Cc}\right)-D_{2}\left(V_{Cc}\right)=0$$
(5)$$D_{1}^{2}\left(1-D_{1}\right)={\left(1-D-D_{1}\right)}^{2}\left(D-D_{1}\right)$$
(6)$$V_{Cc}=\frac{D-D_{1}}{1-D}V_{in}$$
(7)$$V_{DS}\left(Q_{M}\;and\;Q_{C}\right)=\frac{1-D_{1}}{1-D}V_{in}$$
(8)$$D_{1}=\frac{\left(1+D\right)-\sqrt{1-2D+5D^{2}}}{2}$$

In other words, it is possible to reduce the voltage stress of main switch *Q_M_* and clamp switch *Q_C_* through the addition of the additional switch *Q_T_* and simultaneous switching with the main switch *Q_M_*.

## 3. Characteristics and Voltage Stress Control Circuits of the Proposed Converter

The three-switch active-clamp forward converter has a possibility of reduced efficiency due to switching loss and conduction loss occurring in the additional switch with high current stress of the additional diode. In order to minimize this disadvantage, by delaying the turn-off time of *Q_T_* compared to the turn-off time of *Q_M_*, the voltage stress of *Q_M_* and *Q_C_* is slightly increased, but current stress of additional diodes and the turn-off switching loss of additional switch can be reduced as shown in Figure 4. Additionally, it is possible to reduce the peak current and to improve the zero-voltage switching performance by lowering the DC bias of the transformer magnetizing current. That is, as the duty ratio *D_T_* of the additional switch *Q_T_* is increased, it becomes closer to the conventional active-clamp forward converter, and if the *Q_T_* is continuously maintained in the turn-on state, it eventually operates the same as the conventional active-clamp forward converter.

By varying duty ratio *D_T_* of *Q_T_* flexibly according to input/output voltages and output load conditions, its operation can maximize the advantages of the conventional active-clamp forward converter and three-switch active-clamp forward converter. In the case of the conventional active-clamp forward converter, the voltage stress of *Q_M_* and *Q_C_* can be expressed by Equation (3), but in the case of the proposed converter, by reflecting the case when duty of the additional switch *Q_T_* can be increased compared to the conventional, *D_1_* is changed as Equation (9).
(9)$$D_{1}=\frac{\left(1-D^{2}\right)-\sqrt{{\left(1-D^{2}\right)}^{2}-4D\left(1-D\right)\left(1-2D_{T}+2DD_{T}-D^{2}\right)}}{2\left(1-D\right)}$$

Eventually, according to the purpose of the three-switch active-clamp forward converter to reduce voltage stress, a control circuit that can vary *D_T_* within the range that limits the maximum allowable voltage stress is configured as shown in Figure 5. In the case of a control circuit, the maximum voltage can be set by considering the voltage rating of *Q_M_* and *Q_C_*. As the duty ratio *D_T_* are increased, the voltage stress of the main switch *Q_M_* and the clamp switch *Q_C_* increases. By taking advantage of this, it is a method that allows the circuit to operate while maintaining the desired voltage stress by increasing the duty ratio *D_T_*, as much as there is voltage margin.

## 4. Operation Principles

### 4.1. Mode Analysis

The proposed converter operates either as an active-clamp forward converter or as a three-switch active-clamp forward converter, depending on the duty ratio of the additional switch *Q_T_*. That is, in a load requiring a small duty ratio of the main switch, it will operate close to an active-clamp forward converter, and in a load requiring a large duty ratio of the main switch, it will operate close to a three-switch active-clamp forward converter. Therefore, the mode analysis of the proposed converter focuses on the middle operation area between the above two topologies, and describes by dividing into four operation modes, as shown in Figure 6. In addition, the key waveforms of the proposed converter are as shown in Figure 7.

Mode 1 (t_1_~t_2_): The main switch *Q_M_*, and additional switch *Q_T_* are turned on at the same time. Therefore, an input voltage is applied to the transformer, *i_LK_* rises in a positive direction, energy is transferred to the secondary side, *D_S1_* is conducted, and an output is transferred to the load.

Mode 2 (t_2_~t_3_): The main switch *Q_M_* turns off, and the additional switch *Q_T_* maintains turn-on state. Thus, the output capacitance of main switch *Q_M_* is charged and the output capacitance of clamp switch *Q_C_* is discharged. At this time, the transformer current *i_LK_* flows through the additional switch *Q_T_* and body-diode of clamp switch *Q_C_*, so *i_LK_* decreases slowly. As the clamp switch *Q_C_*, which reached the zero voltage, turns on, *i_LK_* conducts through the channel of the clamp switch *Q_C_* and the additional switch *Q_T_*. The secondary diodes *D_S1_* and *D_S2_* commutation each other.

Mode 3 (t_3_~t_4_): When the additional switch *Q_T_* is turned off, the output capacitance of additional switch *Q_T_* is charged by the input voltage, so the input voltage and the clamp capacitor voltage are applied to the transformer together (*V_in_ + V_Cc_*). At the same time, *i_LK_* flows to the input side through the additional diode *D_A_*, the channel of the clamp switch *Q_C_*, and the clamp capacitor *C_C_*. Accordingly, when *i_LK_* reaches 0, the voltage charged in the output capacitance of additional switch *Q_T_* is discharged, and the voltage of clamp capacitor *V_Cc_* is applies to the transformer. In turn, energy cannot be transferred to the secondary side and *D_S2_* conducts by energy stored in the output inductor.

Mode 4 (t_4_~t_5_): As the output capacitance of additional switch *Q_T_* is discharged, *i_LK_* is conducted through the channel of the clamp switch *Q_C_* and body-diode of additional switch *Q_T_*. Additionally, the transformer current *i_LK_* increases in the negative direction, as the clamp switch *Q_C_* turns off, the output capacitance of clamp switch *Q_C_* is charged and the output capacitance of main switch *Q_M_* is discharged by *i_Lk_* to achieve the zero-voltage switching. After this mode ends, mode 1 starts and the following operations repeat.

### 4.2. ZVS Analysis

In the case of the conventional active-clamp forward converter, the ZVS condition of the clamp switch *Q_C_* is relatively easily achieved by the falling *i_LK_* at the transformer reset operation, and the ZVS condition of main switch *Q_M_* is achieved, by *i_LK_*, which is been increased the negative direction peak. However, the range of achieving the ZVS condition of the main switch *Q_M_* is relatively narrow due to the asymmetric current waveform as the offset of *i_LK_* is biased in the positive direction as a whole. Additionally, in the case of the three-switch active-clamp forward converter, the main switch *Q_M_* and clamp switch *Q_C_* can achieve the ZVS condition from each peak current of *i_LK_* before each switching operation, similar to the conventional active-clamp forward converter. For the additional switch *Q_T_*, the ZVS condition can be achieved by the resonance of *L_M_* and *Coss._QT_* during the transformer reset operation. However, main switch *Q_M_* has the asymmetrical feature of the transformer having a slightly elevated offset in the positive direction compared to the conventional active-clamp forward converter. Due to this, main switch *Q_M_* has a narrow ZVS compared to the conventional active-clamp forward converter range.

The proposed converter has a similar ZVS range as it features both topologies above. However, the primary side-switching element has lower voltage stress than the conventional active-clamp forward converter under the same transformer turn ratio condition. Therefore, it allows to use the relatively superior performance MOSFET which leads to a lower *Coss* of the MOSFET, and directly affects the ZVS condition achievement as in Equations (10)–(13).
(10)$$\frac{1}{2}L_{K}i_{LM}^{2}\left(t_{5}\right)\geq \frac{1}{2}\left(C_{oss,QM}+C_{oss,QC}\right){\left(V_{in}+V_{Cc}\right)}^{2}$$
(11)$$i_{LM}\left(t_{5}\right)=\frac{-V_{in}}{2\left(L_{M}+L_{K}\right)}DT_{s}$$
(12)$$\frac{1}{2}L_{K}i_{LK}^{2}\left(t_{2}\right)\geq \frac{1}{2}\left(C_{oss,QM}+C_{oss,QC}\right){\left(V_{in}\right)}^{2}$$
(13)$$i_{LK}\left(t_{2}\right)=\frac{N_{S}}{N_{P}}\left(\frac{\left(\frac{N_{S}}{N_{P}}\right)V_{in}\left(1-D\right)}{2L_{o}}+\frac{V_{in}}{2\left(L_{M}+L_{K}\right)}\right)DT_{S}+\frac{DV_{in}}{R}$$
where *i_LM_(t_5_)* means the negative peak current of magnetizing inductance*(L_M_)*, *i_LK_(t_2_)* means the positive peak current of leakage inductance*(L_K_)*, and *C_OSS_* means the output capacitance of each MOSFET. In addition, for the proposed converter, by operating the additional switch *Q_T_* before turning-on the main switch *Q_M_*, the body-diode conduction interval of *Q_T_* is minimized and the MOSFET channel conduction interval is increased to minimize conduction losses. The discharge time of the switch *Q_T_* is expressed as *t_zvs_,_QT_* in Equation (14). *D_L_* is described as much as duty is ahead. For this reason *D_L_* increases within a equation *D_L_ ≤ D_2_ − t_zvs_,_QT_*.
(14)$$t_{ZVS,QT}\geq \sqrt{\frac{\left(C_{oss,QT}+C_{j,DA}\right)\left(L_{M}+L_{K}\right)V_{in}}{V_{Cc}}}$$

## 5. Design and Simulation for 800 W Charger

Prior to the experiment, the battery charger of the 800 W DC/DC converter for electric kickboard is verified and analyzed through simulation. 36 V (5 Ah) and 48 V (5 Ah) batteries are mainly applied to electric kickboards, and the charging current is up to 15 A. When to charge the battery, CC-CV control is required according to the characteristics of the battery, and the voltage/current control strategy is simplified as shown in Figure 8. The maximum power of 800 W is when the 48 V battery is charged, the output voltage is 53.3 V and the charging current is 15 A, and when the 36 V battery is charged, the maximum power is 600 W at the battery voltage of 40 V. Therefore, the performance is compared and verified for four conditions: output 27 V 15 A, 42 V 4.5 A, 53.3 V 15 A, and 56 V 4.5 A. The design of the main parameters is based on the voltage stress limit of 550 V (85% of the rated voltage of the 650 V FET) of the *Q_M_* of the three-switch active-clamp forward converter to use the 600 V/650 V rated voltage MOSFET with excellent performance index. It is designed as shown in Table 1, and input/output specifications are also shown.

Since the voltage stress of the additional switch *Q_T_* is fixed to input voltage, MOSFET which has 600 V rated voltage and a low R_ds(on)_ is selected. Because the maximum voltage stress of *D_S1_* is 200 V, the diode with 300 V rated voltage and excellent reverse recovery characteristics is selected for the secondary diode. The circuit is designed to enter the DCM mode in 4.5 A load condition, which is the point of slow charging when the charging rate exceeds 80% to reduce the switching loss. Thus, output inductance is selected as 27.8 uH. In addition, before the experiment, the effectiveness is verified through efficiency and loss analysis based on the simulation results as shown in Figure 9.

Through the analysis of simulation results, the proposed voltage-stress-controllable three-switch active-clamp has less overall loss compared to the conventional active-clamp forward converter can be seen. In particular, the proposed converter has significantly reduced switching loss compared to the conventional converter, despite the increase in the absolute number of elements due to the added additional switch *Q_T_*, additional diode *D_A_*. In the fast-charging condition based on 36 V battery charging 27 V, 15 A (405 W), the switching loss is reduced by about 6.12 W, leads to an efficiency increase of about 1.5% can be expected, and in slow charging condition of 42 V, 4.5 A (189 W), is reduced by about 5.06 W, and an efficiency increase of about 2.7% can be expected. In addition, based on 48 V battery charging, under fast charging conditions of 53.3 V, 15 A (799.5 W), switching loss is reduced by about 7.6 W, so an efficiency increase of about 1% can be expected, and in slow charging condition of 56 V 4.5 A (252 W), by about 6.17 W switching loss is reduced, so about 2.5% efficiency increase can be expected. Under all load conditions, the average efficiency increase is expected to be approximately 1.9% due to reduced switching losses, and the result of efficiency is shown in Figure 10. Accordingly, the proposed converter can be expected to lead an overall increase in efficiency compared to the conventional converter.

## 6. Experimental Results

Based on the simulation verification results, in order to verify the validity from actual experiments, a voltage stress controllable three-switch active-clamp forward converter which is capable of charging both 36 V (5 Ah) and 48 V (5 Ah) batteries for electric kickboards has been implemented. In addition, the experimental conditions are shown in Figure 11, and the equipment used is shown in Table 2.

For a 36 V battery, 27 V 15 A load condition, the waveform is as shown in Figure 12. Under the condition of 36 V battery fast charging (27 V 15 A), even if the additional switch *Q_T_* is continuously turned on, the maximum voltage of the set main switch does not exceed 550 V. Therefore, the proposed converter operates the same as the conventional active clamp forward converter. In this case, since the proposed converter employs lower voltage rated FET with smaller Rds(on), it can reduce conduction loss compared to the conventional converter.

For a 36 V battery, 42 V 4.5 A load condition, the waveform is as shown in Figure 13. Under the condition of slow charging of 36 V battery (42 V 4.5 A), the additional switch *Q_T_* continues to conduct as it does not exceed the maximum voltage set by the MOSFET as in fast charging, and it operates same as the conventional active clamp forward converter. Therefore, as described above, compared to the conventional converter, the proposed converter can reduce conduction loss of MOSFET, and it can be confirmed that the switching loss can be reduced by operating in DCM mode during slow charging as intended in the design stage.

For a 48 V battery, 53.3 V 15 A load condition, the waveform is as shown in Figure 14. In the case of 48 V battery fast charging mode 53.3 V 15 A, it can be seen that the voltage stress of the main switch *Q_M_* reaches 550 V, the maximum range sets as shown in Figure 14b. Therefore, it can be confirmed that the duty *D_T_* of the additional switch on the level that maintains the maximum voltage stress is increased and the duty of the additional switch *Q_T_* is increased forward by the lead duty (*D_L_*), which guarantees the ZVS of the additional switch. Therefore, since the duty of the main switch *Q_M_* is higher than the previous 36 V battery condition, the duty *D_T_* of the additional switch is slightly increased. Additionally, as shown in Figure 14d, when the additional switch *Q_T_* is turned off, the voltage waveform of the secondary-side rectifier *D_S2_* is additionally charged as much as *VC_C_* and then discharged.

For a 48 V battery, 56 V 4.5 A load condition, the waveform is as shown in Figure 15, The slow charging condition of the 48 V battery (56 V 4.5 A) is also the maximum voltage tracking control that maintains the set maximum voltage of 550 V as shown in Figure 15b, and the duty *D_T_* is increased accordingly. Switching loss can be reduced by operating in DCM mode at light load in the same manner as in the previous 36 V battery slow charging condition (42 V 4.5 A). Additionally, similar to the fast charging mode, the duty is advanced within the range that guarantees the ZVS of the additional switch *Q_T_*.

It is noteworthy that the proposed converter has a problem which the voltage of clamp capacitor is additionally charged in the rectifier on the secondary side by mounting the additional switch *Q_T_* and results an increase in voltage stress. However, when operating in DCM mode under light load as shown in Figure 15c,d, the voltage stress applied to the secondary rectifier has a similar level to the conventional one.

Therefore, it is possible to comparatively cancel out the critical disadvantage high voltage stress that occurs in the rectifier on the secondary side due to the advantage, gained by adding MOSFET on the primary side, which is a characteristic of the three-switch forward converter.

As a result of the experiment, the efficiency under each condition is shown in Figure 16. Similar to the simulation analysis, it is confirmed that the power loss occurring in each condition is reduced, leading to a relative increase in efficiency. In particular, by operating in DCM mode at the light load, the relative efficiency of the proposed converter could be further increased. Thus, the overall efficiency of the proposed converter could be improved by more than 2% compared to the conventional converter in the light load.

## 7. Conclusions

In this paper, an independent gate control technique is proposed to maximize the efficiency and improve the flexibility of the three-switch active-clamp forward converter. The proposed converter reduces the turn-off switching loss of the two additional switching elements by delaying the duty of the additional switch, and it can reduce the current stress of the added additional diode. In addition, by advancing the duty of the additional switch, the ZVS of the additional switch is ensured and the conduction loss is further reduced. Furthermore, a control method to follow the maximum voltage stress is applied to the proposed circuit by utilizing the rise in voltage stress, which is a disadvantage caused by the turn-off delay. As a result, the proposed converter has the advantages of both the conventional active-clamp forward converter and the three-switch active-clamp forward converter, and it can be flexibly used in wide input dc/dc applications such as dc/dc power supply and battery charging.

## Figures and Tables

**Figure 1 micromachines-14-00035-f001:**
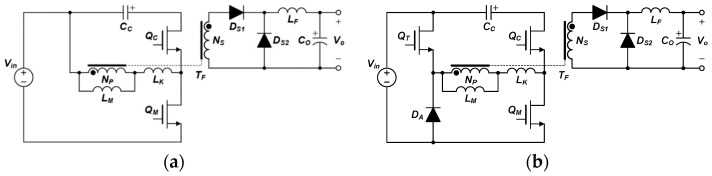
(**a**) Conventional active-clamp forward converter; (**b**) three-switch active-clamp forward converter.

**Figure 2 micromachines-14-00035-f002:**
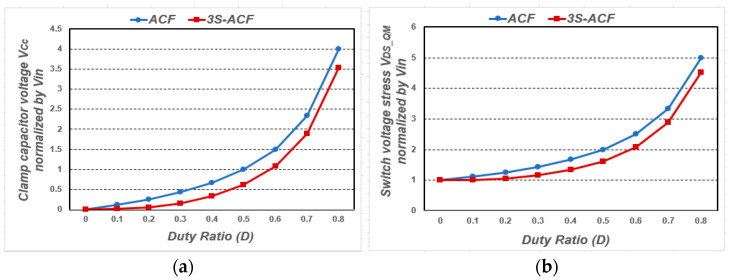
Voltage stress normalized by *V_in_* (**a**) clamp capacitor; (**b**) main switch *Q_M_*.

**Figure 3 micromachines-14-00035-f003:**
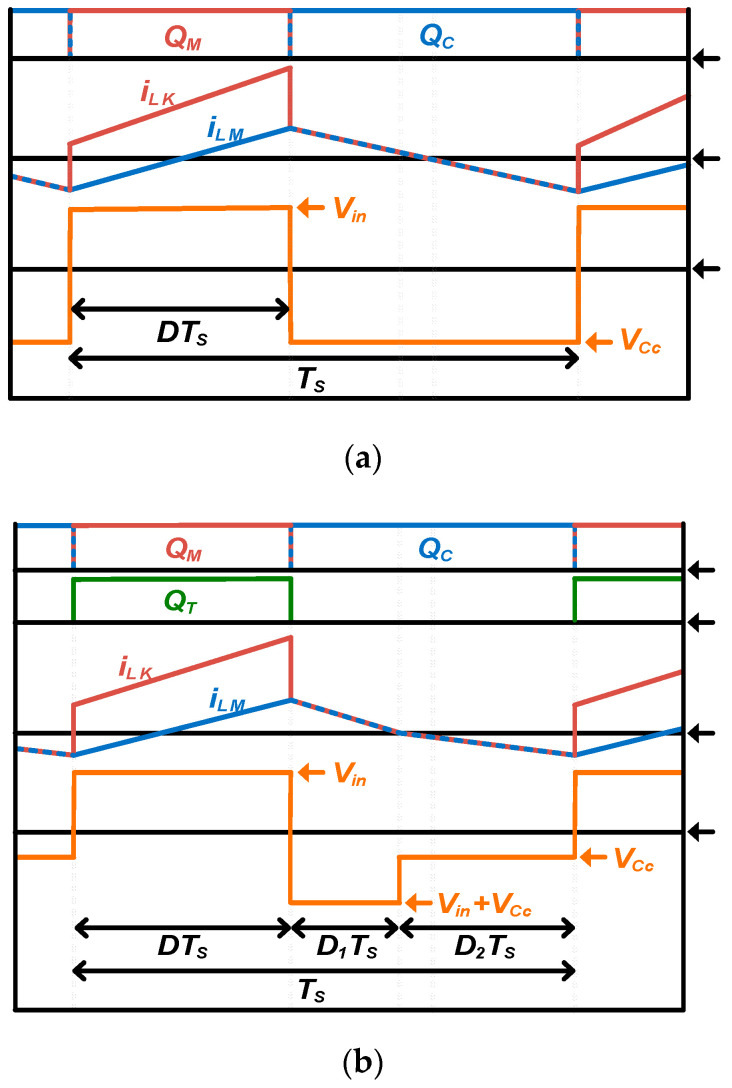
Simplified transformer waveform (**a**) active-clamp forward converter; (**b**) three-switch active- clamp forward converter.

**Figure 4 micromachines-14-00035-f004:**
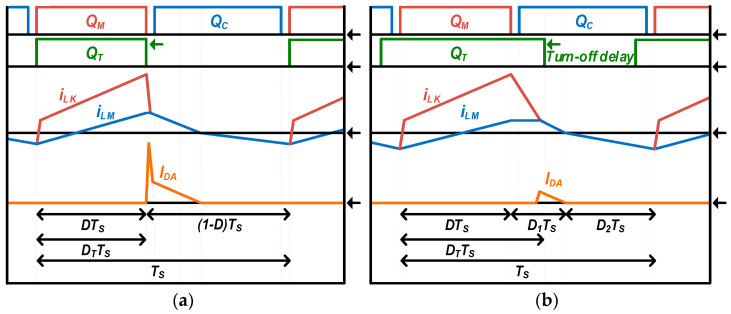
Simplified additional diode current waveform: (**a**) three-switch active-clamp forward converter; (**b**) proposed converter.

**Figure 5 micromachines-14-00035-f005:**
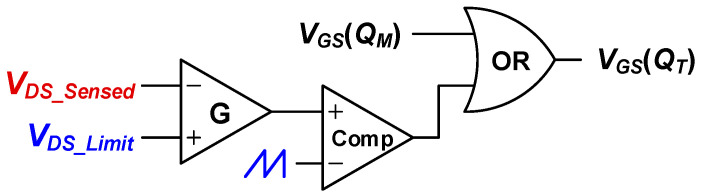
Voltage stress control circuit applied to the proposed converter.

**Figure 6 micromachines-14-00035-f006:**
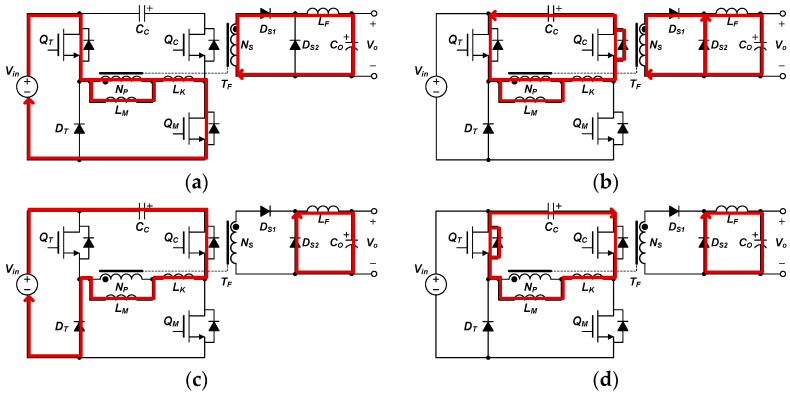
Operation modes of voltage controllable three-switch active-clamp forward converter: (**a**) Mode 1 (t_1_~t_2_), (**b**) Mode 2 (t_2_~t_3_), (**c**) Mode 3 (t_3_~t_4_), (**d**) Mode 4 (t_4_~t_5_).

**Figure 7 micromachines-14-00035-f007:**
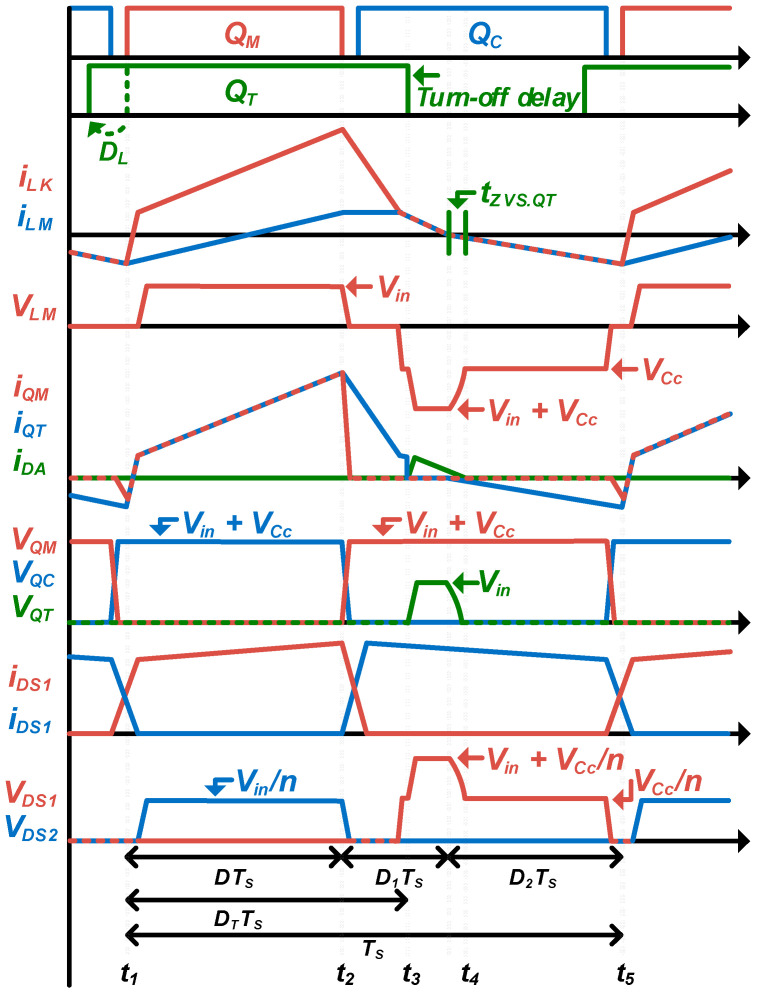
Key waveforms of the proposed converter.

**Figure 8 micromachines-14-00035-f008:**
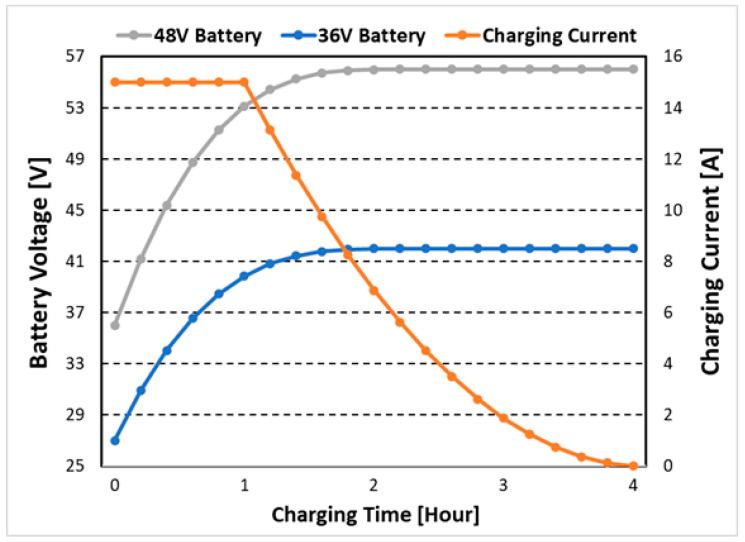
CC-CV control of 36 V and 48 V batteries for electric kickboard.

**Figure 9 micromachines-14-00035-f009:**
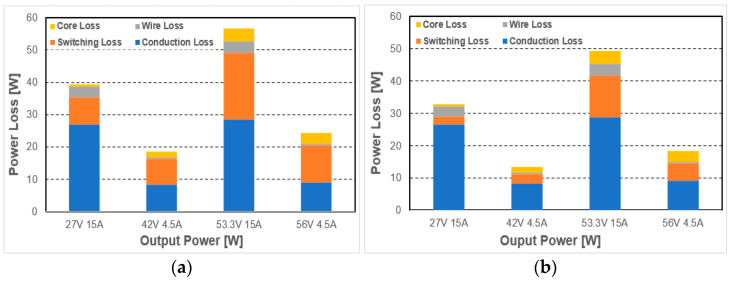
Simulation loss analysis: (**a**) conventional active-clamp forward converter; (**b**) proposed active-clamp forward converter.

**Figure 10 micromachines-14-00035-f010:**
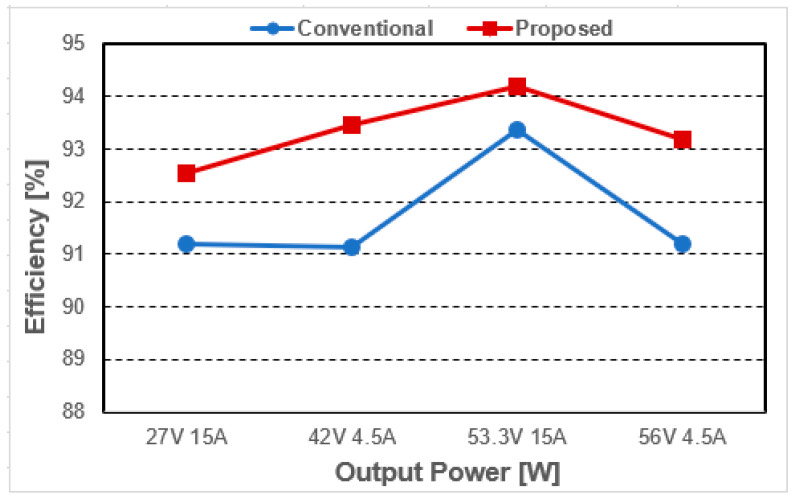
Simulation efficiency.

**Figure 11 micromachines-14-00035-f011:**
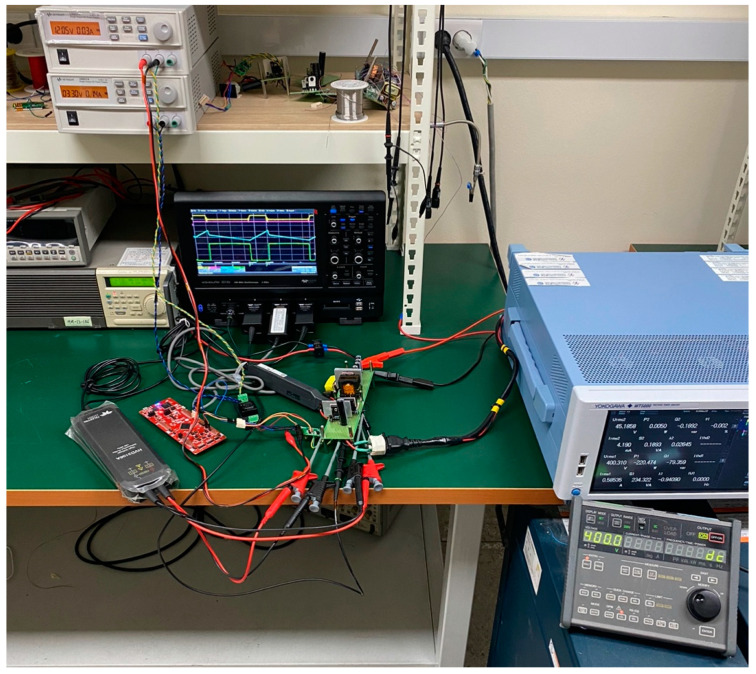
Experimental condition.

**Figure 12 micromachines-14-00035-f012:**
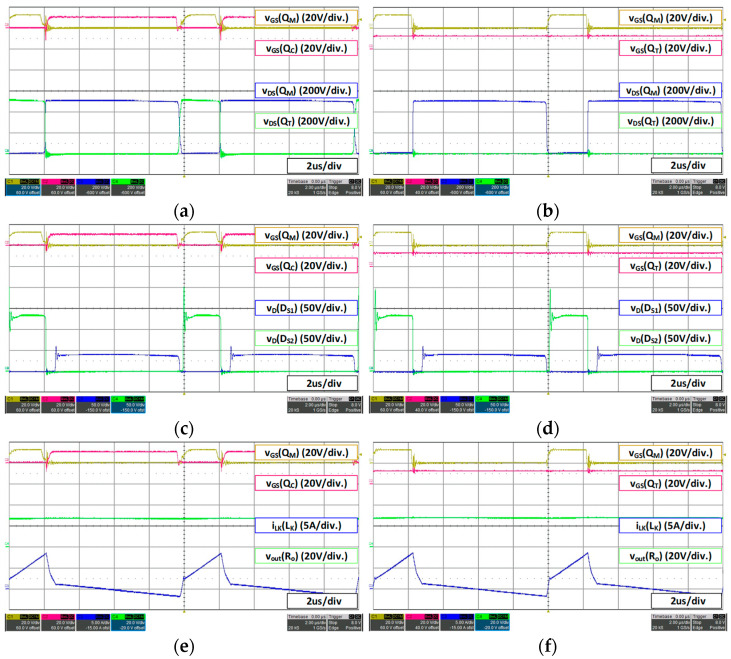
36 V battery (27 V 15 A) load waveform: (**a**) conventional converter MOSFET voltage stress; (**b**) proposed converter MOSFET voltage stress; (**c**) conventional converter secondary rectifier voltage stress; (**d**) proposed converter secondary rectifier voltage stress; (**e**) conventional converter *V_out_*; (**f**) proposed converter *V_out_*.

**Figure 13 micromachines-14-00035-f013:**
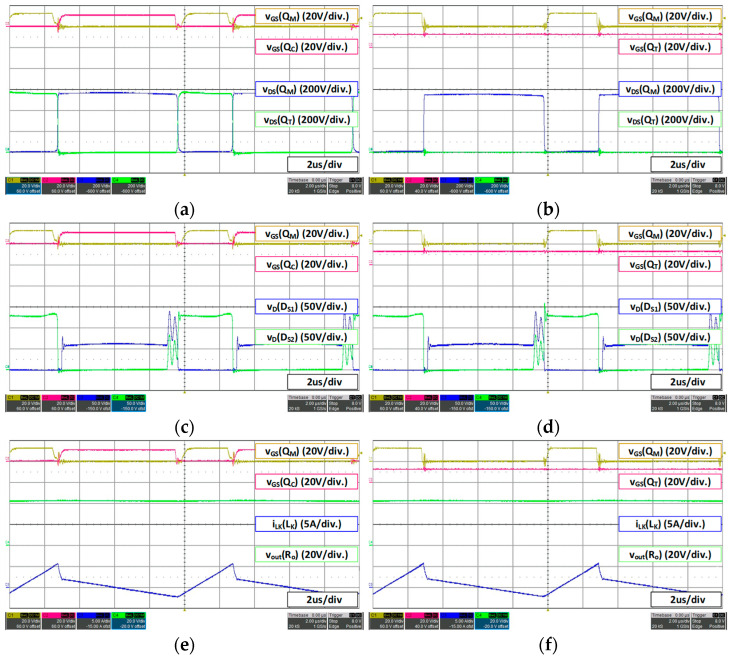
36 V battery (42 V 4.5 A) load waveform: (**a**) conventional converter MOSFET voltage stress; (**b**) proposed converter MOSFET voltage stress; (**c**) conventional converter secondary rectifier voltage stress; (**d**) proposed converter secondary rectifier voltage stress; (**e**) conventional converter *V_out_*; (**f**) proposed converter *V_out_*.

**Figure 14 micromachines-14-00035-f014:**
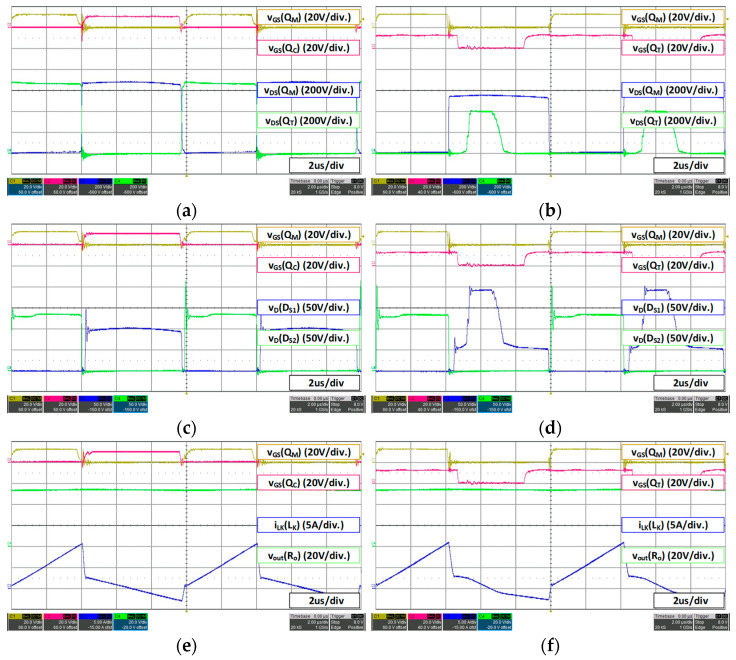
48 V battery (53.3 V 15 A) load waveform: (**a**) conventional converter MOSFET voltage stress; (**b**) proposed converter MOSFET voltage stress; (**c**) conventional converter secondary rectifier voltage stress; (**d**) proposed converter secondary rectifier voltage stress; (**e**) conventional converter *V_out_*; (**f**) proposed converter *V_out_*.

**Figure 15 micromachines-14-00035-f015:**
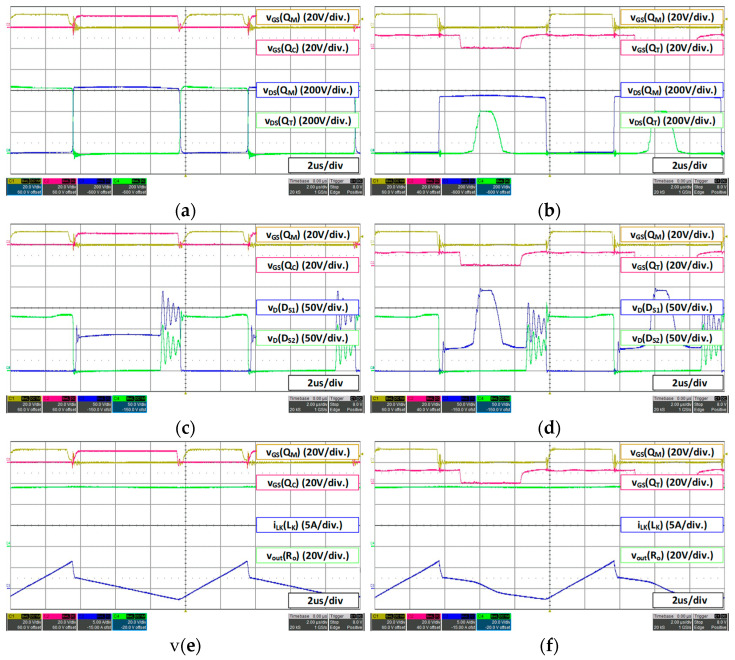
48 V battery (56 V 4.5 A) load waveform: (**a**) conventional converter MOSFET voltage stress; (**b**) proposed converter MOSFET voltage stress; (**c**) conventional converter secondary rectifier voltage stress; (**d**) proposed converter secondary rectifier voltage stress; (**e**) conventional converter *V_out_*; (**f**) proposed converter *V_out_*.

**Figure 16 micromachines-14-00035-f016:**
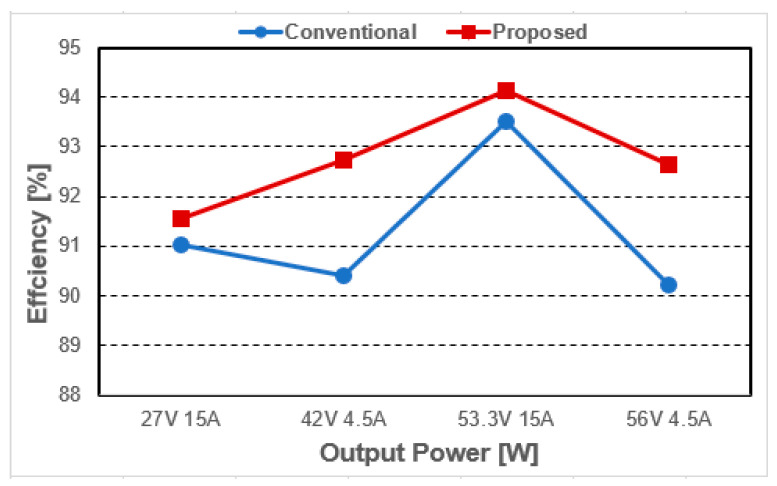
Experimental efficiency.

**Table 1 micromachines-14-00035-t001:** Experimental parameters.

Input voltage, *V_in_*	385 VDC Nominal (380~405 VDC)	
Output voltage, *V_out_*	27 V~56 V (36 V & 48 V Nominal)	
Maximum output current *I_out max_*	15 A	
Switching Frequency	100 KHz	
Topology type	Three-switch active-clamp forward converter	Active-clamp forward converter
Main switch, *Q_M_*	IPW65R090CFD7	IPW90R120C3
Clamp switch, *Q_C_*	IPP65R190CFD7	IPP90R340C3
Additional switch, *Q_T_*	IPW60R018CFD7	None
Transformer, *T_F_*	PQ3230 (22:8 *LM*: 253 uH, *LK*: 14.1 uH)	
Secondary side rectifier diode	LQA30T300(300 V, 1.4 V_F_)	
Output inductor	CH270060 (L_O_: 27.8 uH)	
Output capacitor	80 V 47 uF, 4 EA	

**Table 2 micromachines-14-00035-t002:** Experimental equipment.

DC power source	ES 2000B, ES 2000S
Precision Power Analyzer	WT 5000
Electronic Load	PLZ1003WH
Oscilloscope	WAVESURFER 3014Z
Current Probe	CP030A
Voltage Probe	PP009-1
Differential Probe	HVD3106A

## Data Availability

Not applicable.

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
