# Peer review of "Drain-Source Voltage-Controllable Three-Switch Active-Clamp Forward Converter for Wide Input/Output Voltage Applications"

_micromachines, 2022, doi:10.3390/mi14010035_

Round 1
Reviewer 1 Report
A three-switch DC/DC converter has been presented based on conventional active-clamp forward converter in the manuscript and its operation has been analyzed. Authors must answer and apply the following comments,
1- Few papers have been reviewed in the introduction. More previous existing works must be discussed and compared in this section (Minimum totally 20 papers) for example, the following paper should be compared:
[*] “Lyapunov-Based Control Strategy for a Single-Input Dual-Output Three-Level DC/DC Converter”
2- Why did not author consider a controller for this converter? What benefits and drawbacks a controller can make for the authors’ converter? Please compare with a DC/DC converter involved with a simple controller in the following in the introduction.
[**] “Feedback–feedforward control technique with a comprehensive mathematical analysis for single‐input dual‐output three‐level dc–dc converter”
3- It seems the authors emphasised on using the converter as a battery charger, whereas the converter is not bidirectional DC/DC converter. What are the advantages of this converter compared to another bidirectional DC/DC converters?
4- One of the advantages of the converter is using the three switches that has led to the voltage stress reduction. In two references above [*] and [**], the DC/DC converter has four switches and two outputs leading to less voltage stress and more outputs compared to the author’s converter. Please discuss these points in the introduction section.
5- Can the authors please clarify how their converter reached wide Inputs and outputs?
6- The results are good. Please provide a figure from experimental setup and the related technical information.
Author Response
I would like to thank editor-in-chief and all reviewers for their time and reviewing this paper.
I faithfully answered the comments mentioned by all reviewers, and the answers below were also reflected in the revised paper. Also, there are many changes in Section 6 (Experimental results). I would like to inform you that detailed experimental waveforms that were previously omitted have been added.

Reviewer 2 Report
The authors propose a three-switch active-clamp forward converter topology with controllable drain-source voltage that aims to reduce the reverse voltage stress suffered by semiconductors in the classical two-switch topology.
The proposal concludes with a functional laboratory test.
The analysis of the stressing voltage to be controlled is comparatively analyzed between the classical and the proposed topology.
The principles and modes of operation of the proposal are also developed.
The results and conclusions are based on the simulation and experimental data obtained.
The bibliography, being apparently sufficient, presents a high percentage of citations older than 5 years (~85%), which leads me to wonder if this is due to the fact that this line of work is too mature or this proposal intends to revive or solve a problem that the industry eventually has reminded. Anyway, it is only to satisfy this reviewer's curiosity, not a drawback in this research.
Having said that, and with the sole purpose of improving the quality of the paper, the following suggestions are made:
- Lines 49, 67, 107, 138, 139, 214, 263, 264, 268, 269 and section References: Please, check if the content fits the margins.
- Line 117: What is QA in figure 5?
- Lines 138 and 139: Depicting the inverse diode of the FETs would help in understanding the modes of operation.
- Line 142, Figure 7: Please differentiate between variable names and the value they reach.
- Line 149 and following: What is meant by ‘charge/charging’ refers to the storage of energy in electromagnetic form; what do you mean when you indicate that a switch is ‘charged’? That it has to withstand a reverse voltage, that it goes into load by blocking voltage, and that it goes into load by blocking voltage, isn’t it? Please clarify the meaning of ' charge/charging' in the paragraph.
- Line 156: There seems to be an unnecessary carriage return.
- Line 160: The energy can be ‘energy’ (generically) or 'the energy', but 'an energy'... Sounds strange.
- Line 189: After the period in line 188 (the enumeration of equations does not count as part of the text) should start in capital letters.
- Line 195: Any parameter (DL, for instance) should be described or defined in its first use.
- Line 214, Figure 9: The height of subfigures (a) and (b) are not the same, consider matching them.
- Before the References, MDPI usually employs a final Section on Authors' Contributions, Funding, Acknowledgements and Conflict of Interest Declaration which has been omitted, any special reason for this?
I hope that the review of these suggestions by the authors and their consideration, if any, will serve for the final publication of this paper to be favourably considered.
Author Response

(The authors gave the same response as above.)
